# Predictors of Hyperuricemia after Kidney Transplantation: Association with Graft Function

**DOI:** 10.3390/medicina56030095

**Published:** 2020-02-25

**Authors:** Inese Folkmane, Lilian Tzivian, Elizabete Folkmane, Elina Valdmane, Viktorija Kuzema, Aivars Petersons

**Affiliations:** 1Faculty of Medicine, Univeristy of Latvia, LV-1004 Riga, Latvia; liliana.tz@gmail.com (L.T.); folkmane.elizabete@gmail.com (E.F.); elina.glaudina@gmail.com (E.V.); 2Pauls Stradins Clinical University Hospital, Centre of Nephrology of Latvia, LV-1002 Riga, Latvia; viktorija.kuzema@stradini.lv (V.K.); aivars.petersons@stradini.lv (A.P.); 3Holon Institute of Technology, Holon 5810201, Israel; 4St. Bonifatius Hospital, 49808 Lingen (Ems), Germany; 5Faculty of medicine, Riga Stradins University, LV-1007 Riga, Latvia

**Keywords:** hyperuricemia, uric acid, kidney transplantation, glomerular filtration rate

## Abstract

*Background and objectives*: In kidney transplant recipients (KTR), hyperuricemia (HU) is a commonly-observed phenomenon, due to calcineurin inhibitors and reduced kidney graft function. Factors predicting HU, and its association with graft function, remains equivocal. *Materials and Methods*: We conducted a retrospective longitudinal study to assess factors associated with HU in KTR, and to determine risk factors associated with graft function, measured as glomerular filtration rate (GFR). Moreover, GFR > 60 mL/min/1.73 m^2^ was considered normal. HU was defined as a serum uric acid level of > 416 μmol/L (4.70 mg/dL) in men and >357 μmol/L (4.04 mg/dL) in women, or xanthine-oxidase inhibitor use. We built multiple logistic regression models to assess predictors of HU in KTR, as well as the association of demographic, clinical, and biochemical parameters of patients with normal GFR after a three-year follow-up. We investigated the effect modification of this association with HU. *Results*: There were 144 patients (mean age 46.6 ± 13.9), with 42.4% of them having HU. Predictors of HU in KTR were the presence of cystic diseases (OR = 9.68 (3.13; 29.9)), the use of diuretics (OR = 4.23 (1.51; 11.9)), and the male gender (OR = 2.45 (1.07; 5.56)). Being a younger age, of female gender, with a normal BMI, and the absence of diuretic medications increased the possibility of normal GFR. HU was the effect modifier of the association between demographic, clinical, and biochemical factors and a normal GFR. *Conclusions*: Factors associated with HU in KTR: Presence of cystic diseases, diuretic use, and male gender. HU was the effect modifier of the association of demographic, clinical, and biochemical factors to GFR.

## 1. Introduction

Traditionally, hyperuricemia (HU) is associated with clinical consequences when saturating concentrations result in symptomatic urate or uric acid crystal deposition—gout, acute or chronic uric acid nephropathy, and urolithiasis [1], which makes it more complicated to assess the role of asymptomatic HU in the development and progression of chronic kidney disease; the main route for elimination of uric acid (UA) is via the kidneys. It is accepted that almost consistently, the fall in glomerular filtration rate (GFR) will also lead to an increase in serum UA. Epidemiological studies show that up to 70% of patients with chronic kidney disease (grades IV and V) have HU [2]. However, the association of HU and chronic kidney disease remains unclear: Different studies investigating the association between HU and chronic kidney disease [2,3,4,5,6] either discuss it is an independent risk factor of chronic kidney disease, or as its consequence.

Recent experimental studies demonstrate many deleterious effects of UA on vascular cells, resulting in smooth muscle cell proliferation and inflammation: this occurs through increased production of growth factors (platelet-derived growth factor, PDGF), vasoconstrictive substances (cyclooxygenase-2 induced thromboxane and angiotensin II) and proinflammatory molecules (C-reactive protein and monocyte chemoattractant protein) [7,8,9]. HU can induce oxidative stress and endothelial dysfunction, resulting in the development of systemic and glomerular hypertension; this can lead to reduced renal blood flow and gradual decline in kidney function [3,10]. UA may also exert a direct effect on renal tubular cells, and induce epithelial-mesenchymal transition, with further generation of matrix-producing fibroblasts and kidney fibrosis [11].

HU has a particularly high prevalence in the KTR population, or 30% to 84% [12]. In addition to traditional HU risk factors, such as decreased GFR, use of diuretics, male gender, diabetes mellitus, hypercalcemia, and higher body weight, these patients are also exposed to non-traditional risk factors such as treatment with calcineurin inhibitors, especially cyclosporine A [12,13,14,15]. The pathogenetic mechanisms of kidney injury induced by HU in KTR are similar to those in chronic kidney disease and may affect kidney graft function. Early experimental animal models show that histological changes induced by HU were identical to chronic graft nephropathy. HU can exacerbate cyclosporine A toxic effect, suggesting it could be one of the modifiable, donor-independent potential risk factors of chronic kidney transplant dysfunction [7,8,16].

However, studies investigating the association of HU with kidney transplant function and the effect of UA-lowering therapy on kidney transplant function are scarce. Very few studies investigated factors related to HU itself. In this study, there were two specific aims: (1) To investigate demographic, clinical, and biochemical factors that could be associated with HU, and (2) to assess the role of HU as an effect measure modifier in the association between these factors and as a function of kidney transplant.

## 2. Materials and Methods

We performed a longitudinal retrospective study, using patient records with kidney transplantation from 01.01.2009 until 31.12.2011 in the Latvian Transplantation Centre. Only patients with data available for more than 12 months were included in the analysis, patients with a gout diagnosis were excluded. This study was approved by the Ethical Committee for Scientific Research of the University of Latvia (approved in 07.10.2015, Nr 47/2015). We also assessed demographic parameters of patients (age, gender, body-mass index—BMI in the first year after kidney transplant further grouped patients as <30 kg/m^2^ for normal weight, and ≥30 kg/m^2^ as overweight), clinical parameters (hypertension, diabetes mellitus, cystic diseases), biochemical parameters (glucose, UA, level of serum creatinine, and serum cholesterol levels), and use of medications that could affect HU after kidney transplantation (immunosuppressive therapy, angiotensin-converting enzyme (ACE) inhibitors/angiotensin receptor blocker, xanthine oxidase inhibitors, and/or diuretics). Hypertension was defined as systolic blood pressure > 140 mmHg and diastolic blood pressure > 90 mmHg, or regular use of blood pressure lowering medications. Presence of diabetes mellitus was defined as the presence of diagnosis in patient records, or a glucose level of ≥126 mg/dl, or use of antidiabetic medications. HU was defined as the level of uric acid in blood >416 μmol/l (4.70 mg/dL) for men or 357 μmol/L (4.04 mg/dL) for women, or the use of xanthine oxidase inhibitors. Dyslipidemia was defined as the level of total serum cholesterol >200 mg/dl or the regular use of antilipidemic medications. All patients received triple immunosuppressive therapy—calcineurin inhibitors (cyclosporine A or tacrolimus), antiproliferative agents (mycofenolate mofetil or azathioprine), and prednisone. Data were collected about the use of ACE inhibitors/angiotensin receptor blockers, diuretics, and xanthine oxidase inhibitors (allopurinol/febuxostat) as well.

For the subsequent analysis, we calculated the function of the kidney transplant after the third year follow-up, with GFR (ml/min/1.73 m^2^), according to the Cockroft-Gault equation:
$$GFR=\frac{\left(\mathrm{age}-140\right)*Body\;mass\;\left(kg\right)}{0.81*\mathrm{Serum}\;\mathrm{Creatinine}\;\left(\frac{\mu \mathrm{mol}}{l}\right)}\times (0.81\mathrm{if}\;\mathrm{female}).$$

We further divided GFR at the threshold of 60 mL/min/1.73 m^2^ to assess normality of kidney transplant function, with GFR higher than 60 mL/min/1.73 m^2^ considered as normal. 

Statistical analysis was performed using the Statistical Program for the Social Sciences (SPSS) software, 23^rd^ version (IBM Corp., Released 2015. IBM SPSS Statistics for Windows, Version 23.0. Armonk, NY, USA). A *p*-value of less than 0.05 was considered statistically significant. Differences between patients both with and without HU in their demographic, clinical, and biochemical parameters, and with the use of medications were assessed using Chi-square/Exact Fisher tests for group variables or the t-test for continuous variables, as all continuous variables in this study were normally distributed. 

We performed two-stage multivariate analysis: 

*Stage 1*. Multiple logistic regression model was built to assess demographic, clinical, and biochemical factors associated with HU. 

*Stage 2*: The multiple logistic regression model investigated the association of demographic, clinical, and biochemical factors with GFR after a three-year follow-up. The adjustment set for the main model was based on variables univariately associated with GFR, with a significance level at 0.1, and other clinically meaningful variables. The full adjustment set consisted of age, gender, BMI (<30 or ≥30), hypertension, dyslipidemia, diabetes mellitus, use of ACE inhibitors/angiotensin receptor blockers, and diuretics. 

*Effect modification*: We investigated the possible effect measure modification by HU for the association between predictors and GFR. 

*Sensitivity analysis*: We performed a sensitivity analysis to evaluate the possible role of HU as a predictor of GFR, using the model built for Stage 2. As such, we also adjusted the main model for HU. 

## 3. Results

Of 168 patient records who had kidney transplantation performed, 24 patients were excluded from the study due to unavailable data, or death less than twelve months after transplantation. Therefore, the analysis was performed for 144 eligible patients, of whom 61 had HU. Most participants (84%) were younger than 60 years, with a normal (less than 30 kg/m^2^) BMI (79.9%). We found the same proportion of men and women in the study sample (males, 49.3%). Groups with and without HU differed on dyslipidemia and the presence of cystic diseases, use of ACE inhibitors/angiotensin receptor blockers, and use of diuretics (Table 1). All clinical and biological parameters, as well as the use of medication, were higher in patients with HU (Table 1), while less patients with normal GFR had HU (35.2% vs. 45.0%) however, this difference was not significant. 

*Stage 1*. From all demographic, clinical, and biochemical parameters of included patients, we used a multiple logistic regression for prediction of HU in KTR, the major predictor being the presence of cystic diseases (OR = 9.68 (3.13; 29.9)) followed by use of diuretics (and a four-fold higher risk of HU in those who used diuretics), or those of the male gender (Table 2). No other factors were associated with HU in KTR. 

*Stage 2*. Factors associated with normal GFR after a three-year follow-up (GFR > 60) were a younger age (OR = 0.95 (0.92; 0.99)), the female gender (OR = 0.31 (0.13; 0.74)), BMI less than 30 kg/m^2^ (OR = 0.17 (0.05; 0.54)), and the absence of diuretic medications (OR = 0.15 (0.03; 0.65)) (Table 3). The proposed logistic regression explained 29.2% of variance for GFR by all factors included in the model. 

*Effect modification*: We observed an effect modification from HU for the association observed in the main model. For patients without HU, factors associated with normal GFR were female gender (OR = 0.13 (0.03; 0.57)) and normal BMI (OR = 0.01 (0.01; 0.19)), although for patients with HU, the only factor significantly associated with normal GFR was younger age (OR = 0.93 (0.87; 0.99)) (Table 4). For patients with HU, the explained variation in GFR was very high—or 82.2%—which as an explanation for GFR variation in patients without HU, was extremely different: Nagelkerke adjusted R^2^ = 4.0%. 

*Sensitivity analysis*: We did not observe HU to be predictive of normal GFR (OR = 1.20 (0.49; 3.00)) (Appendix A).

## 4. Discussion

In this study, we observed that three major factors associated with HU in KTR included: presence of cystic diseases, use of diuretics and male gender. HU was the effect modifier of the association of demographic, clinical, and biochemical factors with GFR, as a better explanation of variation in GFR for those with HU. 

According to our research, the presence of cystic diseases was a significant predictor of HU after kidney transplantation. To our knowledge, there are no studies that analysed cystic kidney disease as a risk factor for HU in KTR. Future studies must be performed to confirm this finding. 

According to many epidemiological studies, the main diseases considered to be associated with a risk of HU are diabetes, metabolic syndrome, hypertension, and cardiovascular diseases [13,14,15,16,17]. Of all cystic kidney diseases, HU is more frequently associated with autosomal dominant tubulointerstitial kidney disease (ADTKD), as well as autosomal dominant polycystic kidney disease (ADPKD) [18,19].

The physiological mechanisms of HU in these two diseases are completely different. ADTKD is a group of genetic kidney diseases that cause progressive loss of kidney function, thereby resulting in end stage renal disease (ESRD) in the third through seventh decade of life [18]. HU and gout are mainly associated with ADTKD—uromodulin (UMOD) and ADTKD—renin (REN) genetic forms [20], sometimes starting in the teenage years but usually preceding development of renal failure. The possible pathophysiological mechanisms of HU are reduced activity of the Na^+^, K^+^, and 2Cl^-^ cotransporter; this results from decreased levels of uromodulin in the UMOD genetic form, or renin deficiency that leads to aldosterone deficiency in the REN genetic form, which subsequently decreases sodium and chloride reabsorption, in turn leading to a volume depletion which may promote proximal reabsorption of UA [21,22]. 

Also, ADPKD is the most common inherited kidney disorder, known to affect all ethnic groups at a prevalence of 1:400–1:1000 live births [19]. This is caused by mutations in one of two genes, PKD1 (chromosome region 16p13.3; ∼85% of cases) and PKD2 (4q21; ∼15% of cases) [23]. The pathogenetic mechanism of HU in ADPKD might be explained by altered tubular membrane transport process resulting in impaired renal urate handling and homeostasis [24]. Also, hemodynamic changes, such as a decrease in renal blood flow with preserved GFR, can result in an increased filtration fraction with a consequent increase in peritubular oncotic pressure; a rise in sodium and uric acid reabsorption has been detected in patients with ADPKD [25]. Recently, a genome-wide association of studies identified multiple genetic loci connected to kidney disease-related traits, including uric acid levels. It has been shown that genetic variability around the PKD2 locus could contribute to serum uric acid concentrations in different populations [26,27]. 

According to the results of our study, we would like to emphasize that regular testing of HU in patients with an underlying diagnosis of cystic kidney disease is especially important after kidney transplantation in order to prevent symptomatic HU. It should be noted that with haemodialysis, patients often have normal or even lower uric acid levels [28], but after kidney transplantation, immunosuppressive therapy may exert a permissive effect in this regard, delaying the overt symptoms of gout, or recognition of the need for treatment [12]. In the context of transplantation, genetic testing is important for healthy family members of those with cystic disease, who are willing to serve as potential kidney transplant donors [29].

Our study also found that the use of diuretics was independently associated with higher risk for HU. In the general population, diuretics are one of the most important causes of secondary HU. Loop diuretics and thiazide diuretics interact with renal organic anion transporters (OAT), entering the proximal tubular cell from the blood side via OAT1 and OAT3 transporters may be considered as competitive substrates of uric acid [30]. Moreover, diuretics reduce uric acid excretion, presumably by causing mild volume depletion with enhancement of proximal tubular reabsorption [31]. Similarly, the association between diuretic use and HU in KTR has been reported by Bandukwal et al. [32] and Eyupoglu et al. [33]. This is of practical importance, as the of elevated blood pressure of KTR has been reported to be as high as 85% [34], but diuretics have remained the cornerstone of antihypertensive treatment since their introduction in the 1960s. Considering a very high HU and gout risk of diuretics, practitioners should carefully evaluate indications for the use of diuretics in the KTR population. 

In the current study, we observed that the male gender to be a significant predictor of HU. Gender is a common nonmodifiable risk factor for HU and gout, which is the most severe complication of HU in the general adult population. Men have a greater risk of developing gout than women in all age groups, although the sex ratio tends to equalize with advancing age [35]. In a managed care population in the USA, the male/female ratio in patients with gout was 4:1 for those younger than 65, and 3:1 for those older than 65 [35]. Data are similar to Malheiro et al. [13] and Numakura et al. [14] studies. In contrast, some other studies found that HU was more frequent in the younger female gender, including KTR [36] and [37]. Gender differences may be explained by an enhancement of renal urate excretion in women of childbearing age, due to the effects of estrogenic compounds, which likely reduce the number of active renal urate transporters, and resulting in less renal tubular uric acid reabsorption and in increased urate clearance [38]. 

Our study confirmed that the male gender is an independent risk factor for post-transplant HU. Thus, this could be the group of patients who need more intensive monitoring of UA concentration and as such, initiation of UA-lowering treatment is necessary to protect the kidney graft function.

The results of our study did not indicate HU as an independent risk factor for graft function. We demonstrated that in certain clinical situations HU may affect renal transplant function along with some other risk factors. Controversial to our results, some studies revealed possible association between increased serum uric acid concentration and transplant failure. For example, in FAVORIT study with 3512 patients followed up for 3.9 years [39] uric acid concentrations were not independently associated with mortality or transplant failure. This can be explained by a strong association between concentration on UA and some other risk factors, for example, with GFR> However, in some newer studies that show the effectiveness of treatment for HU in KTR with Febuxostat [40,41], concerns for drug interactions are less compared to the CNIs-Allopurinol interaction. Future longitudinal studies are needed to assess an effectiveness of treatment of HU with Febuxostat and its effect on clinical, and specifically allograft outcomes. 

### 4.1. Limitations of the Study

This study had several limitations. First, a small number of patients can reduce the power of a study, so more studies are needed to assess possible predictors of HU, as well as its role in KTR. In addition, a possibility of human mistakes whilst using electronic data bases should be taken into account. A second and major limitation of this study was the absence of electronic registers in Latvia, which makes more human mistakes possible. As one of the limitations of this study should be mentioned a lack of possibility to assess the number of patients with HU, that did not undergo KTR. In that case, we cannot be fully sure that some of observed associations will remain correct for patients without KTR. For example, lack of association with cyclosporine that was observed in this study may be not proved by studies that include participants with and without KTR. Finally, 24 patients who had been lost to follow-up could be those with an abnormal GFR, and this loss would be meaningful for such a small sample size.

### 4.2. Strengths of the Study

To our knowledge, this is the first study that investigates sociodemographic factors that predict HU, as well as the role of HU in KTR. Without a doubt, this study adds to the global evidence about KTR patients and provides understanding about the necessity for therapy in cases of asymptomatic HU. The novelty of the study is the fact that hyperuricemia after kidney transplantation is an effect modifying factor. Up to date, there is a broad discussion in literature in the effect of hyperuricemia on graft function. We demonstrated that in certain clinical situations (demographic, medication, etc.), hyperuricemia may affect graft function. We also identified cystic kidney disease as a significant risk factor for hyperuricemia after kidney transplantation. Our study is in line with studies that exist in the field. The fact that we can reproduce results of studies performed in completely different conditions and in completely different populations provides additional validation of results of studies by other authors affirming previously observed results.

## 5. Conclusions

In this study, factors associated with HU in KTR were presence of cystic diseases, use of diuretics, and male gender. HU might be seen as the effect modifier of the association of demographic, clinical, and biochemical factors with GFR. According to this study, it can be proposed for patients with abnormal GFR to have special care even in the case of asymptomatic HU.

## Figures and Tables

**Table 1 medicina-56-00095-t001:** Demographic, clinical, and biochemical patient parameters, and use of medications by hyperuricemia (HU) groups.

Variable	Patients without HU, N = 83	Patients with HU, N = 61	*p* Value
Demographic parameters			
Age (years), mean ± SD	44.3 ± 14.0	48.9 ± 13.8	0.09
Male gender, N (%)	38 (45.8)	33 (54.0)	0.21
BMI (kg/m^2^), mean ± SD	24.9 ± 4.7	25.4 ± 5.3	0.08
BMI > 30 (kg/m^2^), N (%)	15 (18.1)	14 (23.0)	0.30
Clinical and biochemical parameters			
GFR at the 3^rd^ year > 60, N (%)	36 (45.0)	19 (35.2)	0.17
Hypertension, N (%)	62 (74.7)	51 (83.6)	0.14
Diabetes mellitus, N (%)	11 (13.2)	7 (11.5)	0.48
Dyslipidemia, N (%)	40 (48.2)	38 (62.3)	0.05
Presence of cystic diseases, N (%)	6 (7.2)	17 (27.9)	< 0.01
Use of medications, N (%)			
Cyclosporine A, N (%)	46 (55.4)	38 (62.3)	0.26
Tacrolimus, N (%)	30 (36.1)	23 (37.7)	0.49
MMF, N (%)	80 (96.4)	56 (91.8)	0.21
Prednisone, N (%)	51 (61.4)	38 (62.3)	0.48
AKE-I/ARB, N (%)	43 (51.8)	43 (70.5)	0.02
Diuretics, N (%)	9 (10.8)	15 (24.6)	0.02

**Table 2 medicina-56-00095-t002:** Factors associated with HU in kidney transplant recipients (KTR) *.

Variable	OR	95% CI	*p* Value
Age	1.01	0.97; 1.04	0.72
Male gender	2.45	1.07; 5.56	0.03
BMI ≥ 30	1.15	0.40; 3.29	0.79
Hypertension	0.72	0.23; 2.23	0.57
Dyslipidemia	2.27	0.96; 5.36	0.07
Diabetes mellitus	0.64	0.20; 2.01	0.44
Presence of cystic diseases	9.68	3.13; 29.9	< 0.01
Use of AKE-I/ARB	1.11	0.71; 1.72	0.65
Use of diuretics	4.23	1.51; 11.9	0.01

* Nagelkerke adjusted R^2^ = 0.24.

**Table 3 medicina-56-00095-t003:** Factors associated with normal glomerular filtration rate (GFR) after a three-year follow-up (GFR > 60) *.

Variable	OR	95% CI	*p* Value
Age	0.95	0.92; 0.99	< 0.01
Male gender	0.31	0.13; 0.74	< 0.01
BMI ≥ 30	0.17	0.05; 0.54	< 0.01
Hypertension	1.59	0.51; 4.95	0.43
Dyslipidaemia	1.11	0.46; 2.70	0.82
Diabetes mellitus	1.60	0.48; 5.30	0.45
Use of AKE-I/ARB	1.10	0.73; 1.66	0.65
Use of diuretics	0.15	0.03; 0.65	0.01

* Nagelkerke adjusted R^2^ = 0.29.

**Table 4 medicina-56-00095-t004:** Effect measure modification for the association of GFR by HU *.

	Patients without HU			Patients with HU		
Variable	OR	95% CI	*p* value	OR	95% CI	*p* value
Age	0.96	0.91; 1.01	0.13	0.93	0.87; 0.99	0.04
Male gender	0.13	0.03; 0.57	< 0.01	0.57	0.14; 2.30	0.43
BMI ≥ 30	0.01	0.01; 0.19	< 0.01	1.00	0.18; 5.62	0.99
Hypertension	2.94	0.56; 15.49	0.20	2.00	0.13; 31.98	0.62
Dyslipidemia	1.50	0.39; 5.83	0.56	0.31	0.05; 2.13	0.23
Diabetes mellitus	2.41	0.37; 15.84	0.36	1.10	0.15; 8.12	0.93
Use of AKE-I/ARB	1.24	0.77; 2.01	0.38	1.13	0.12; 10.42	0.91
Use of diuretics	0.01	0.00; 0.00	1.00	0.43	0.07; 2.70	0.37

* Nagelkerke adjusted R^2^ for patients without HU = 0.04; for patients with HU = 0.82.

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
