# Peer review of "Predictors of Hyperuricemia after Kidney Transplantation: Association with Graft Function"

_medicina, 2020, doi:10.3390/medicina56030095_

Round 1
Reviewer 1 Report
The concept of the study is good. Hyperuricemia is a common finding in the renal transplant recipients. As authors stated, the etiology is most likely multifactorial. Authors found male gender, cystic diseases, use of diuretics as risk factors. However, the paper needs extensive english language, syntax and grammar edition; I would suggest that the authors use english-language editing service to help increase the visibility of this paper.
Also, suggest adding conventional units for serum uric acid since most North American readers are unfamiliar with the SI units that the authors stated in the paper.
Author Response
Predictors of hyperuricemia after kidney transplantation: association with graft function
Response to Reviewer 1
Dear reviewer,
Thank you for your constructive comments which helped us to focus and improve the manuscript. As a result of revising a manuscript according to your commentaries, we made substantial changes to our previous work. Please see our answers on your comments. Changes in the manuscript body are made accordingly.
I would suggest that the authors use English-language editing service to help increase the visibility of this paper.
Dear Reviewer, according to your suggestion we sent the manuscript to English editing service. Please see attached a certificate from the American Manuscript Editors.
Also, suggest adding conventional units for serum uric acid since most North American readers are unfamiliar with the SI units that the authors stated in the paper.
Dear Reviewer, thank you for this comment. We added a calculation for hyperuricemia (in mg/dL) in abstract and in methods sections. The limit of HU is a serum uric acid level of > 416 μmol/L (4.70 mg/dL) in men and >357 μmol/L (4.04 mg/dL) in women.

Reviewer 2 Report
In this study, authors investigated the association between kidney transplant and hyperuricemia. Authors found presence of cystic disease, male gender and use of anti-diuretics potential risk factors. I have following concerns.
Hyperuricemia and gout are common problems in renal transplant patients. Use of cyclosporine is clearly been associated with hyperuricemia as it lowers the urinary clearance of uric acid. However, in this report authors did not find association of cyclosporine use in hyperuricemic patients as compared to non-HA patients with cyclosporine.
The important question is, how many patients had HA before renal transplant?
Sample size is low to draw authors conclusion and likely more human mistakes in electronic registry is noted
Author Response
Predictors of hyperuricemia after kidney transplantation: association with graft function
Response to Reviewer 2
Dear reviewer,
Thank you for your constructive comments which helped us to focus and improve the manuscript. As a result of revising a manuscript according to your commentaries, we made substantial changes to our previous work. Please see our answers on your comments. Changes in the manuscript body have been made accordingly.
Hyperuricemia and gout are common problems in renal transplant patients. Use of cyclosporine is clearly been associated with hyperuricemia as it lowers the urinary clearance of uric acid. However, in this report authors did not find association of cyclosporine use in hyperuricemic patients as compared to non-HA patients with cyclosporine. The important question is, how many patients had HA before renal transplant?
Dear reviewer, thank you for this comment. Our study population included only patients with kidney transplant recipients (KRT), of them 42.4% of them had hyperuricemia (HU). The aim of this study was to investigate the role of HU only in patients with KRT, therefore, we have no possibility to assess the number of patients with HU before renal transplant. However, we gratefully acknowledge your comment and added this as a limitation of the study. The new text is:
As one of the limitations of this study should be mentioned a lack of possibility to assess the number of patients with HU, that did not undergo KTR. In that case, we cannot be fully sure that some of observed associations will remain correct for patients without KTR. For example, lack of association with cyclosporine that was observed in this study may be not proved by studies that include participants with and without KTR.
Sample size is low to draw authors conclusion and likely more human mistakes in electronic registry is noted
We acknowledge a small sample size in this study and fully agree with you about an extreme strength of conclusions that we made. Therefore, we changed the section of conclusions according to your suggestions. We pointed out as well a possibility of human mistake in limitations section of the manuscript. The new text is:
In limitation of the study section: In addition, a possibility of human mistakes whilst using electronic data bases should be taken into account.
In conclusion section: In this study, factors associated with HU in KTR were presence of cystic diseases, use of diuretics, and male gender. HU might be seen as the effect modifier of the association of demographic, clinical, and biochemical factors with GFR. According to this study, it can be proposed for patients with abnormal GFR to have special care even in the case of asymptomatic HU.
Round 2
Reviewer 2 Report
Thank you for answering my questions. Please see the following published papers showing association of HU with male gender, use of diuretics in renal transplant patients. Therefore, it is hard to find novelty in the present manuscript by Folkmane I.et.al.
Clive DM. Renal transplant-associated hyperuricemia and gout. J Am Soc Nephrol. 2000;11:974–979. [PubMed] [Google Scholar] Kalantar E, Khalili N, Hossieni MS, Rostami Z, Einollahi B. Hyperuricemia after renal transplantation. Transplant Proc. 2011;43:584–585. [PubMed] [Google Scholar] Malheiro J, Almeida M, Fonseca I, Martins LS, Pedroso S, Dias L, Henriques AC, Cabrita A. Hyperuricemia in adult renal allograft recipients: prevalence and predictors. Transplant Proc. 2012;44:2369–2372. [PubMed] [Google Scholar] Kim KM, Kim SS, Han DJ, Yang WS, Park JS, Park SK. Hyperuricemia in kidney transplant recipients with intact graft function. Transplant Proc. 2010;42:3562–3567. [PubMed] [Google Scholar] Numakura K, Satoh S, Tsuchiya N, Saito M, Maita S, Obara T, Tsuruta H, Inoue T, Narita S, Horikawa Y, et al. Hyperuricemia at 1 year after renal transplantation, its prevalence, associated factors, and graft survival. Transplantation. 2012;94:145–151. [PubMed] [Google Scholar]
Author Response
Response to reviewer – round 2
The comment of the reviewer was: Please see the following published papers showing association of HU with male gender, use of diuretics in renal transplant patients. Therefore, it is hard to find novelty in the present manuscript by Folkmane I.et.al
Dear reviewer, thank you for this additional comment. Certainly, up to date there is a lot of literature investigate the same thematic. Moreover, all studies presented in your comment are those that we mentioned as references for our study. Nevertheless, we are sure that our study is valuated for this field of knowledge. We acknowledge the fact that we should show more precisely a novelty of this study. To our knowledge, the novelty of the study is the fact that hyperuricemia after kidney transplantation is an effect modifying factor. Up to date, there is a broad discussion in literature in the effect of hyperuricemia on graft function. We demonstrated that in certain clinical situations (demographic, medication, etc.), hyperuricemia may affect graft function. We also identified cystic kidney disease as a significant risk factor for hyperuricemia after kidney transplantation. We added this comment (underlined) to discussion section.
In addition, repeatability of studies is mentioned in many articles (e.g. Ioannidis 2005[1]; Moonesinghe et al 2007[2]) as one of the major goals of contemporary epidemiology and medicine. In particular, in the study by Meerpohl et al. (2011)[3] is mentioned, that the willingness to produce a new knowledge should be accomplished by reproducing of the previous studies to achieve their reliability, and stress to publish makes the newly produced knowledge doubtful and reduces public trust in science. Therefore, even in case that our study partly reproduces previously cited studies, we see it rather not as a weakness but as a strength of it. We added this comment to discussion section. The new text is:
Our study is in line with studies that exist in the field. The fact that we can reproduce results of studies performed in completely different conditions and in completely different populations provides additional validation of results of studies by other authors affirming previously observed results.
[1] Ioannidis JPA. Why most published research findings are false? PloS Medicine. 2005; 2(8):e124
[2] Moonesinghe R, Khoury MJ, Cecile A, Janssens JW. Most published research findings are false – but a little replication goes a long way. PloS Medicine. 2007; 4(2):e28
[3] Meerpohl JJ, Wolff RF, Antes G, von Elm E. Are pediatric Open Access journals promoting good publication practice? An analysis of author instructions. BMC Pediatr. 2011;11:27. doi: 10.1186/1471-2431-11-27.
